# Autoimmune Disease Classification Based on PubMed Text Mining

**DOI:** 10.3390/jcm11154345

**Published:** 2022-07-26

**Authors:** Hadas Samuels, Malki Malov, Trishna Saha Detroja, Karin Ben Zaken, Naamah Bloch, Meital Gal-Tanamy, Orly Avni, Baruh Polis, Abraham O. Samson

**Affiliations:** 1Azrieli Faculty of Medicine, Bar Ilan University, Safed 1311502, Israel; hadasamuels10@gmail.com (H.S.); malovmalki@gmail.com (M.M.); trishna.saha27@gmail.com (T.S.D.); karinab1992@gmail.com (K.B.Z.); naamah.bloch@biu.ac.il (N.B.); meital.tanamy@biu.ac.il (M.G.-T.); orly.avni@biu.ac.il (O.A.); 2School of Medicine, Yale University, New Haven, CT 06520, USA; baruhpolis@gmail.com

**Keywords:** autoimmune disease, text mining, frequency analysis, taxonomy, classification, list of autoimmune diseases, PubMed, Alzheimer’s disease

## Abstract

Autoimmune diseases (AIDs) are often co-associated, and about 25% of patients with one AID tend to develop other comorbid AIDs. Here, we employ the power of datamining to predict the comorbidity of AIDs based on their normalized co-citation in PubMed. First, we validate our technique in a test dataset using earlier-reported comorbidities of seven knowns AIDs. Notably, the prediction correlates well with comorbidity (R = 0.91) and validates our methodology. Then, we predict the association of 100 AIDs and classify them using principal component analysis. Our results are helpful in classifying AIDs into one of the following systems: (1) gastrointestinal, (2) neuronal, (3) eye, (4) cutaneous, (5) musculoskeletal, (6) kidneys and lungs, (7) cardiovascular, (8) hematopoietic, (9) endocrine, and (10) multiple. Our classification agrees with experimentally based taxonomy and ranks AID according to affected systems and gender. Some AIDs are unclassified and do not associate well with other AIDs. Interestingly, Alzheimer’s disease correlates well with other AIDs such as multiple sclerosis. Finally, our results generate a network classification of autoimmune diseases based on PubMed text mining and help map this medical universe. Our results are expected to assist healthcare workers in diagnosing comorbidity in patients with an autoimmune disease, and to help researchers in identifying common genetic, environmental, and autoimmune mechanisms.

## 1. Introduction

Autoimmune diseases (AIDs) have a multifactorial etiology including genetic and environmental components. The incidence of AIDs is estimated at 3–5% worldwide. Autoimmunity is known to have a genetic component; yet, concordance rates of AIDs in monozygotic twins are incomplete, indicating a multifactorial etiology. The differences in autoimmunity incidence rates in different ethnic groups and geographical locations suggest the involvement of environmental factors such as lifestyle, exposure to infection, and nutrition, among others.

Autoimmune diseases (AIDs) occur when the immune system attacks self-molecules as a result of a breakdown of immunologic tolerance to autoreactive immune cells [1]. These cells promote the secretion of high-affinity autoantibodies that target and react with the “self” molecules [2]. AIDs are sometimes comorbid, and a higher susceptibility to a second AID is regarded as an indication of potential common pathogenic mechanisms among autoimmune diseases [3]. AIDs are infrequent causes of death, yet they do contribute to mortality and to the quality of life, and their classification is an unmet need [4]. This study focuses on identifying a more extensive association by comparing all of the known AIDs according to their co-citation in the literature.

The idea that different AIDs are associated is not new, and ~25% of patients with one AID tend to develop other comorbid AIDs [5]. A population-based study of 12 AIDs highlighted an overall association between autoimmune disorders [6]. The statistical analysis of the association in that study showed that the number of people with more than one AID was significantly higher than the number predicted under the null hypothesis. In another study, there were significantly more cases than expected of ALS associated with a prior diagnosis of asthma, celiac disease, type 1 diabetes, multiple sclerosis, myasthenia gravis, myxedema, polymyositis, Sjögren’s syndrome, systemic lupus erythematosus, and ulcerative colitis [7]. These results are in agreement with the supposition of common pathogenic mechanisms between autoimmune diseases.

Several approaches have been used to classify human AIDs [1,8]. These approaches have been complicated by the lack of major histocompatibility complex and autoantibody association diseases tentatively labelled as AIDs [9]. As such, classifications have led to revised definitions of autoimmunity, yet these fail to define self-directed tissue inflammation, which is not autoimmune [10]. Furthermore, AID classification has been complicated by deviations in age of onset, gender ratio, monozygotic twin concordance, etc. As such, more recent classifications use SNP allele associations [11], environmental associations [12], and, most recently, machine learning [13]. 

Bioinformatics uses the wealth of information to analyze biological data. Text mining and citation counts are often used to identify trends and patterns in medicine [14]. Several studies have used text mining, and notably Bork et al. have captured the phenotypic effects of drugs based on the side effects resources published by the FDA [15]. In another study, Jensen and coworkers have used text mining to associate diseases and genes, and to establish a web-based database named DISEASE [16]. In the past, we have used frequency analysis of PubMed citations to successfully classify antibiotic resistance [17], antimalaria resistance [18], and Alzheimer’s disease comorbidities [19]. Here, we use PubMed citations to document the co-occurrence of AIDs and determine their interrelated taxonomy. First, we validate our methodology in a small subset of AIDs, and then we apply it to classify AIDs.

## 2. Methods

### 2.1. PubMed Count

To mine for AIDs associations, we counted the number of co-citations on PubMed. The association of two AIDs was taken as the number of PubMed citations of the AID pair (e.g., “Lupus” AND “Sjögren”), divided and normalized by the number of PubMed citations for each disease alone (e.g., “Lupus” OR “Sjögren”) according to the equation below, in which AID_1_ and AID_2_ correspond to AID pairs:(1)$$\mathrm{Association}\;\mathrm{value}=\frac{\left[{“\mathrm{AID}}_{1}{”\;\mathrm{AND}\;“\mathrm{AID}}_{2}”\right]}{\left[{“\mathrm{AID}}_{1}{”\;\mathrm{OR}\;“\mathrm{AID}}_{2}”\right]}$$

The nominator of this equation is biased towards highly cited AIDs. To counter this bias, the association is divided and normalized by the denominator, which reduces the association of highly cited AIDs and increases the association of poorly cited AIDs. This equation also assumes random noise, and signal to noise is expected to grow by the square root of the signal, similar to random walk. To further reduce the chance of random co-occurrence, we queried only the title, abstract, and other terms of PubMed citations, but not the full text paper. Finally, to facilitate the laborious process of counting citations and to prevent scribe copy errors, a Python script was used. The script uses the “wget” function and rapidly looped over all AIDs pairs. The search was performed on a standard PC equipped with 16 GB RAM and operating under Microsoft OS. At an internet speed of 40 Mbps, one query took less than a second, and a complete run of the script took ~6 h. 

### 2.2. List of Autoimmune Diseases

To validate our technique, we calculated the association values of a limited dataset of 7 AIDs reported by Cifuentes et al. [3]. The training dataset comprised systemic lupus erythematosus (SLE), Sjögren’s syndrome (SS), type 1 diabetes (T1D), autoimmune thyroid disease (AITD), rheumatoid arthritis (RA), multiple sclerosis (MS), and systemic sclerosis. To evaluate our technique, we calculated the association between 150 AIDs (listed in Appendix A). The initial list of 150 diseases was obtained by downloading several lists of AIDs and merging duplicates. Autoimmune lists were obtained from the Autoimmune Association website (https://autoimmune.org/disease-information, accessed on 12 July 2021), the Autoimmune Institute website (https://www.autoimmuneinstitute.org/resources/autoimmune-disease-list, accessed on 12 July 2021), and the Autoimmune Registry (https://www.autoimmuneregistry.org/the-list, accessed on 12 July 2021). Inadvertently, some of the 150 diseases were not AIDs and were omitted from our classification. Furthermore, temporary diseases were also excluded because they do not fit with the progressive nature of AIDs. Finally, indistinct AIDs with low PubMed citation counts were also omitted. The final list included only 100 AIDs. 

### 2.3. Clustering

To classify AIDs, we use heatmap clustering based on association values. In particular, the ClustVis website was used for visual classification [20]. In addition, Circos software was used for preparing chord diagrams of AIDs using association values.

## 3. Results and Discussion

### 3.1. Validation

To validate our technique, we calculated the association values of seven AIDs. Figure 1 shows the AIDs association matrix of systemic lupus erythematosus (SSE), Sjögren’s syndrome (SS), type 1 diabetes (T1D), autoimmune thyroid disease (AITD), rheumatoid arthritis (RA), multiple sclerosis (MS), and systemic sclerosis (SSc). Figure 1 also shows the association matrix reported by Cifuentes et al. [3] Remarkably, the correlation between these two matrices is high (R = 0.91) and corroborates our predicted association of AIDs [3]. 

Notably, the association values reported in Figure 1 are small. However, if multiplied by 100, then the association values could represent an approximate percent comorbidity. For example, Sjögren and lupus, which are both collagen diseases, are associated in more than 3.5% (0.0358 × 100) of cases. Notably, associations less than 0.00001 are unrealistic, and have not been included in our study.

### 3.2. Association of Autoimmune Diseases

Figure 2 shows a classification of the top co-cited 100 AIDs, discussed in the following paragraph, in light of their association and comorbidity. The AIDs are listed in the order obtained from ClustVis, and remarkably their clustering follows gender, system, onset age, and frequency. Notably, the automatic classification meticulously organizes all comorbid AIDs according to the major system affected. The affected systems include: (1) gastrointestinal, (2) neuronal, (3) eye, (4) cutaneous, (5) musculoskeletal, (6) kidneys and lungs, (7) cardiovascular, (8) hematopoietic, (9) endocrine, and (10) multiple. Interestingly, the clustering also organizes AIDs according to the major gender affected. Remarkably, the gender classification is separated at male and female gonads AIDs, namely, autoimmune orchitis, which affects the testes, and autoimmune oophoritis, which affects the ovaries. Yet, the male/female disease classification cannot single-handedly explain the classification and could arise from other associations (such as collagen, which is expressed more in men than in women). Note that each affected system is listed twice, once for each gender. The patient gender, average age of onset, and frequency in population are detailed in Appendix A and were taken from Bender et al. [21].

Figure 3 shows the autoimmune association as a chord diagram. Several examples are non-trivial and clearly illustrate the need to classify AIDs [4].

Notably, most AIDs are comorbid, as shown in Figure 2, and the top 20 diseases involve collagen tissues, which explains male predominance. For example, eosinophilic esophagitis is an inflammation of the esophagus (collagen type 1). Among others, it is associated with other AIDs such as autoimmune pancreatitis, retroperitoneal fibrosis, asthma [23] and celiac disease [24], as shown in Figure 2 and Figure 3.

Retroperitoneal fibrosis, which is caused by the buildup of inflammatory and fibrous tissue (collagen type 1) in the retroperineum [25], is associated with Hashimoto’s thyroiditis, Graves’ disease, vasculitis, psoriasis, and autoimmune pancreatitis, among others, as shown in Figure 3. These comorbidities involve multiple systems (Figure 2) such as lymph nodes, pancreas, salivary glands, kidneys, bile ducts, and lachrymal glands. 

Castleman disease is a lymphoproliferative condition, which presents either idiopathically, or in association with herpesvirus [1], or a malignant tumor. Notably, Castleman disease and POEMS syndrome are associated (Figure 2 and Figure 3), and the former is a major criterion of diagnosis of the latter [26]. (POEMS syndrome is an acronym of Polyneuropathy, Organomegaly, Endocrinopathy, Myeloma, and Skin changes. Note collagen involvement)

Guillain Barré syndrome occurs upon demyelination of peripheral nerves (collagen type 6), and is often induced by an infection or a drug [27]. Guillain Barré shares clinical similarities with chronic inflammatory demyelinating polyneuropathy [28], and with peripheral neuropathies, such as POEMS, chronic inflammatory demyelinating polyneuropathy, and multifocal motor neuropathy, as shown in Figure 2. Notably, Guillain Barré syndrome and myasthenia gravis occasionally combine [29], as reflected in Figure 3.

Relapsing polychondritis is characterized by recurrent inflammation of cartilaginous tissues (collagen type 2) throughout the body [30]. Figure 2 shows that relapsing polychondritis is associated with scleritis, uveitis, and conjunctivitis, and both relapsing polychondritis and scleritis are associated with autoimmune otorhinolaryngitis [31]. Figure 3 shows that sympathetic ophthalmia, and Vogt Koyanagi Harada disease are also associated with uveitis [32].

Goodpasture’s syndrome occurs through the deposition of autoantibodies in basement membranes (collagen type 4) of kidneys and lungs, eliciting rapidly progressive glomerulonephritis and pulmonary hemorrhage [33]. Figure 2 also shows that fibrosing alveolitis, also known as idiopathic pulmonary fibrosis, is associated with Goodpasture’s syndrome.

Neonatal lupus and congenital heart block can lead to atrioventricular conduction abnormalities diagnosed in utero or within the first month of life [34]. Maternal autoimmune disease is responsible for and presents in the neonatal heart membranes (collagen type 1) [35]. Figure 2 shows that congenital heart block and neonatal lupus are highly associated [36]. Figure 3 shows that fetal congenital heart block is associated with maternal autoimmunity and with lupus [37].

Eosinophilic fasciitis involves inflammation of tissue under the skin and over the muscle, called fascia (collagen type 1), and it tops the list of AIDs associated with collagen and male predominance [38]. 

Next, stiff person syndrome is caused by autoantibodies to glutamic acid decarboxylase (GAD), which lead to muscle weakness. Figure 3 shows GAD spectrum disorders include cerebellar ataxia, autoimmune epilepsy and encephalitis, among others.

Surprisingly, undifferentiated connective tissue (UCTD) is not clustered with other connective tissue diseases, and Figure 2 suggest it shares less in common with them. Differentiating between UCTD and early-stage SLE is important to avoid irreversible target-organ damage [39] 

Cold agglutinin disease is a form of autoimmune hemolytic anemia following exposure to cold temperature. Interestingly, the onset age depends on the climate, and chilly weather precipitates the disease. As such, as seen in Figure 2, it does not associate with other blood and bone marrow autoimmune diseases such as pure red cell aplasia, hypogammaglobulinemia, and agammaglobulinemia, which arise from a bone marrow autoimmune disease [40]. 

Parry Romberg syndrome (PRS) is characterized by the progressive degeneration of tissues of one side of the face, leading to hemifacial atrophy. Figure 3 shows it is comorbid with other AIDs, such as SLE, rheumatoid arthritis, inflammatory bowel disease, ankylosing spondylitis, vitiligo, and thyroid disorders [41]. Notably, autoimmune involvement is absent in most cases of PRS, and Figure 2 suggests that association with the aforementioned AIDs is weak without the presence of antinuclear antibodies [42].

Cogan’s syndrome is an autoimmune inner ear disease, and Figure 2 and Figure 3 show associations with sympathetic ophthalmia and Vogt–Koyanagi–Harada disease [43]. 

Lambert–Eaton myasthenic syndrome is an autoimmune disorder characterized by muscle weakness of the limbs [44]. Notably, one could expect that Lambert–Eaton (33% female predominance) would be associated with myasthenia gravis (75% female predominance), yet this is not the case, and comorbidity of both is unheard of [45].

Susac’s syndrome is a rare autoimmune disease that mainly affects young women. It is characterized by endotheliopathy, which presents as encephalopathy, retinal vasoocclusive disease, and hearing loss [46]. Figure 2 does not classify Susac’s syndrome with other cardiovascular AIDs, perhaps due to its relatively recent AID classification as attested by antinuclear antibodies, or to a different etiology.

Remarkably, our PubMed clustering differentiates between male and female AIDs. Here too, autoimmune oophoritis and autoimmune orchitis are gonad AIDs, and antibodies bind to both testicular and ovarian target antigens during their development [47].

Mucha–Habermann disease (MHD), also known as “Pityriasis lichenoides et varioliformis acuta” is a skin disease characterized by rashes and small lesions. Figure 2 does not classify MHD with other cutaneous AIDs, perhaps due to its male predominance or to a different etiology. In fact, MHD is a spectrum of diseases and often is triggered by infectious agents, or an inflammatory response secondary to T-cell dyscrasia, or an immune complex-mediated hypersensitivity [48]. 

Progesterone dermatitis (PD) is a skin disease due to progesterone toxicity, for example during the menstrual cycle [49]. Both PD and MHD develop in response to an endocrine disbalance or an external toxic stimulus, which explains their association in Figure 2. Notably, MHD affects more men, while PD affects mainly women, and are unlikely comorbid. Finally, without the presence of antinuclear antibodies, their classification as AIDs could be biased.

Graves’ disease is the most common cause of hyperthyroidism. It is an autoimmune disorder with systemic manifestations that primarily affect the heart, skeletal muscle, eyes, skin, bone, and liver. Figure 2 shows that Graves’ disease and Hashimoto’s thyroiditis are highly associated, and both have been reported to coexist in the same individual, reflecting their common autoimmune origin [50]. Hashimoto’s thyroiditis and Graves’ disease are some of the most common autoimmune endocrine diseases. 

Pernicious anemia is commonly caused by deficiency of vitamin B12 (cobalamin). Sometimes, it is also classified as an autoimmune disease, as it is comprised of salient features of autoimmune chronic atrophic gastritis and cobalamin deficiency [51]. Figure 2 classifies it along with other polyglandular syndromes, as it is often comorbid with these, suggesting a potentially common B12 deficiency.

Trivially, polyglandular syndromes (PGS) type 1, 2, and 3 are highly associated (Figure 2). PGS type 1 is an autosomal recessive syndrome due to mutation of the AIRE gene, resulting in hypoparathyroidism, adrenal insufficiency, hypogonadism, vitiligo, candidiasis, and others [52]. PGS type 2 combines Addison’s disease along with autoimmune thyroid disease and/or type 1 diabetes [53]. Less trivially, PGS type 2, is also named Schmidt’s syndrome, which explains the latter association with all PGS and omission from Figure 2, although it also refers to a brainstem syndrome leading to hemiparesis. Finally, PGS type 3 is the combination of autoimmune Hashimoto’s thyroiditis with another organ-specific autoimmune disease, such as diabetes mellitus, pernicious anemia, vitiligo, alopecia, myasthenia gravis, and Sjögren’s syndrome, as shown in Figure 3. The distinction between these is blurred by their association with coinciding AIDs [52]. Notably, PGS are also known as autoimmune polyendocrine syndrome. 

Dermatitis herpetiformis (formerly known as Duhring’s disease) is associated with other skin diseases such as bullous pemphigoid and pemphigus [54], as shown in Figure 2.

Cicatricial pemphigoid (i.e., bullous pemphigoid, mucous membrane pemphigoid), and ocular cicatricial pemphigoid, are different manifestations of the same bullous disease, which is manifested through blisters all over the body [55]. Figure 3 shows that these bullous diseases can coincide with other AIDs, underlining the involvement of an overactive immune system [56]. 

Discoid lupus is a dermatological AID that can lead to rashes, scarring, hair loss, and hyperpigmentation of the skin, which tends to get worse when exposed to sunlight. Figure 2 shows that discoid lupus is highly associated with both lichen planus and lichen sclerosis and often overlap clinically [57]. Figure 3 shows they are also associated with sarcoidosis, erythema nodosum, and pyoderma gangrenosum, underlining a common genetic etiology [58]. 

Vitiligo is associated with several comorbid autoimmune, systemic, and dermatological diseases, primarily thyroid disease, alopecia areata, diabetes mellitus, pernicious anemia, systemic lupus erythematosus, rheumatoid arthritis, Addison’s disease, inflammatory bowel disease, Sjögren’s syndrome, dermatomyositis, scleroderma, psoriasis, and atopic dermatitis, among others (Figure 3) [59]. 

Alopecia areata is characterized by a well-defined non-scarring alopecic patch or patches that can extend to the entire scalp or lead to total body hair loss [60]. Figure 2 clusters it with vitiligo, and they often overlap clinically.

Figure 2 shows a high association between ulcerative colitis and Crohn’s disease [61]. Despite several differences, ulcerative colitis and Crohn’s disease are different grades of the same chronic bowel inflammation [62]. Notably, the differences arise from different gut microbiota, hormonal factors, and to a lesser extent from autoimmune variations [61]. Interestingly, higher rates of comorbid celiac disease, Crohn’s disease, or ulcerative colitis are found in patients with eosinophilic esophagitis than in the general population [63]. 

Primary biliary cirrhosis and primary sclerosing cholangitis are associated in Figure 2. Both are variations of autoimmune hepatitis with cholestatic characteristics such as autoantibody negative autoimmune hepatitis, giant cell hepatitis, primary biliary cholangitis, and primary sclerosing cholangitis [64].

Myositis is an inflammation of muscles responsible for movement. Trivially, myositis, polymyositis, dermatomyositis, and inclusion body myositis are highly associated in Figure 2 [65], are often comorbid, and share autoimmune antibodies [66]. Polymyositis is associated with scleroderma, conferring an increased risk of connective tissue diseases, such as interstitial lung disease, inflammatory joint disease, SLE, and Sjögren’s syndrome (Figure 3).

Scleroderma is another connective tissue disorder, both local and systemic, and can further be classified as limited systemic sclerosis, formerly known as the CREST syndrome (Figure 2), which comprises calcinosis, Raynaud phenomenon, esophageal dysmotility, sclerodactyly, and telangiectasia [67]. Figure 3 also shows a link between systemic sclerosis and biliary cholangitis [68].

Psoriasis and psoriatic arthritis are trivially associated in Figure 2 [69]. Psoriasis also increases the risk of rheumatoid arthritis, and these diseases have similar comorbidity profiles, with overlapping therapeutic options [69]. Likewise, reactive arthritis, juvenile arthritis, and ankylosing spondylitis (i.e., Bechterew’s disease) are also highly associated under the umbrella term spondyloarthritis (Figure 3) [29].

Figure 2 clusters sarcoidosis with predominantly female AIDs affecting multiple systems and could share a common etiology. Sarcoidosis has been linked to infectious organisms such as mycobacterium and cutibacterium, and certain manifestations of sarcoidosis have been linked to specific HLA alleles, but the overall pathogenesis remains uncertain [70].

Primary Sjögren’s syndrome is a systemic AID that is characterized by a triad of symptoms, namely, dryness, pain and fatigue [71]. Figure 2 associates it with rheumatoid arthritis and SLE. 

Rheumatoid arthritis is a chronic inflammatory joint disease, predominantly of autoimmune origin. Rheumatoid arthritis autoantibodies include the rheumatoid factor and antibodies against citrullinated proteins [72].

Systemic lupus erythematosus (SLE) is a multisystem AID with varied clinical manifestations and a complex pathogenesis [72]. SLE is the most comorbid of AIDs (Figure 3), and few diseases are as devastating as lupus [73]. Figure 3 shows that SLE can also accompany arthritis, scleroderma, myositis, inflammatory bowel diseases, and celiac disease [74]. SLE has also been associated with neuronal AIDs such as myasthenia gravis [75]. Figure 2 clusters SLE with Sjögren’s syndrome and despite several differences antiphospholipid antibodies have been identified in both [76]. 

Glomerulonephritis, IgA nephropathy, and amyloidosis are highly and often associated with arthritis, as shown in Figure 2 [77]. 

Next are cardiovascular AIDs, and Figure 2 shows that uveitis and Behçet’s disease are highly linked, and uveitis is an ocular manifestation of Behçet’s disease [78]. Conspicuously, Figure 3 shows that uveitis is associated with ankylosing spondylitis, psoriatic arthritis, arthritis, colitis, sarcoidosis, Behçet’s disease, and Posner–Schlossman syndrome, among others [79]. 

Although, vasculitis is not specifically autoimmune, inflammation of the blood vessels is associated with AIDs. IgA vasculitis (formerly called “Henoch–Schönlein purpura”), and IgG and IgM vasculitis are three types of leukocystoclastic vasculitis, and their high association is trivial [80]. Rarer vasculitises include granulomatosis with polyangiitis, and microscopic polyangiitis. Kawasaki disease and polyarteritis are highly associated cardiovascular AIDs in Figure 2, and together with Takayasu’s arteritis, polyarteritis nodosa, ANCA-associated vasculitis, giant-cell arteritis, they are coronary artery vasculitises [81]. Figure 3 extensively associates vasculitis with sarcoidosis, erythema nodosum, and pyoderma gangrenosum, among others [82,83]. 

Mounting evidence suggest that narcolepsy is a neuronal AID, and it is associated with restless legs syndrome (not an AID and omitted from Figure 2). In fact, the same patients that find themselves unable to move for a few minutes report restless legs syndrome could be triggered [84]. 

Myasthenia gravis manifests as muscle weakness caused by antibodies against the nicotinic acetylcholine receptor. Figure 3 associates it with other AIDs [85]. 

Figure 2 and Figure 3 show that autoimmune encephalomyelitis and multiple sclerosis are associated with chronic inflammatory demyelinating polyneuropathy, multifocal motor neuropathy, CLIPPERS syndrome, neuromyelitis optica spectrum disorders, and tumefactive demyelinating lesions [86]. Likewise, there is an increased risk of dementia in autoimmune hypothyroidism [87]. Of particular interest to the authors is the association of multiple sclerosis and Alzheimer’s disease, and we shall return to this finding in the discussion.

Optic neuritis, neuromyelitis optica, and transverse myelitis are highly associated in Figure 2. In agreement with this observation, transverse myelitis and optic neuritis are both elements for the diagnosis of neuromyelitis optica [88]. Figure 3 shows that transverse myelitis is also associated with other AIDs, such as lupus [89]. 

Neutropenia is clustered with other hematopoietic diseases in Figure 2. Notably, neutropenia is associated with several non AIDs, such as endometriosis, fibromyalgia, and interstitial cystitis, which have been omitted from Figure 2. In turn, fibromyalgia is associated with SLE, Sjögren syndrome, and IBD [90].

Evans syndrome and hemolytic anemia are also comorbid but do not share the same antibodies.

### 3.3. Misfit AIDs

Some AIDs are omitted from Figure 2 because they are not sufficiently distinct or are included in other categories. For example, juvenile myositis is not distinct from myositis, except it affects children. Juvenile arthritis is indistinct from arthritis and is omitted from Figure 2. Adult Still’s disease is included in Still’s disease, and it is also omitted. Perivenous encephalomyelitis is a subcategory of encephalomyelitis, sometimes identified as acute disseminated encephalomyelitis, and is omitted from Figure 2. Likewise, Baló’s disease (or “Baló’s concentric sclerosis”) forms a subcategory of multiple sclerosis and does not receive a distinct status in Figure 2. IgG4-related sclerosing disease is a subcategory of sclerosis and is omitted from Figure 2 [91]. Likewise, linear IgA is a rare autoimmune blistering disease, with linear IgA deposits along the basement membrane zone [92]. It is associated with other bullous diseases and omitted from Figure 2.

Some AIDs are secondary to other AIDs and do not present as primary AIDs. For example, Raynaud’s phenomenon is idiopathic and not known as a primary AID and is omitted from Figure 2 and Figure 3.

### 3.4. Pathogen-Induced Diseases

Some of the potential AIDs are vector-borne, result from pathogen infections, and are not listed in Figure 2. For example, coxsackievirus myocarditis is not an AID, as it is caused by a virus. Chagas disease is also completely out of place; it is not an AID as it is due to Trypanosoma parasitic infection. Additionally, Lyme disease is out of place, as it is caused by a Borrelia bacterium infection. Mooren’s ulcer is a painful type of peripheral ulcerative keratitis and has been associated with microbial infection [93]. Another example is chronic recurrent multifocal osteomyelitis, a painful bone inflammation, and despite its association with multiple AIDs, may be more associated with bacterial infection [94]. Tolosa Hunt syndrome is caused by an idiopathic granulomatous inflammation of the cavernous sinus and is characterized by painful ophthalmoplegia [95]. It does not present with autoantibodies, and as such is not an autoimmune disease. 

Essential mixed cryoglobulinemia is associated with infections, malignancy, and autoimmune diseases, but may be idiopathic [96]. Up to 90% of reported cases are associated with hepatitis C (HCV) infection, and as such it is omitted from Figure 2.

Finally, rheumatic fever is a disease that can affect the heart, joints, brain, and skin if bacteria infections are not treated properly. Rheumatic fever and subacute bacterial endocarditis are thus highly associated. This should not be confounded with rheumatoid arthritis, and it is excluded from Figure 2. 

### 3.5. Tumor-Induced Diseases

Paraneoplastic cerebellar degeneration is caused by cancer and as such is not included in Figure 2.

### 3.6. Toxicity-Induced Diseases

Some AIDs are direct results of xenobiotic environmental exposure, and their sudden onset is related to known causes. For example, autoimmune urticaria (i.e., hives) is often due to chemical exposure. Benign mucosal pemphigoid is mostly associated with external noxious stimuli, unlike the AID mucous membrane pemphigoid [97], and as such it is omitted from Figure 2. 

In addition, some of the names are not necessarily associated with AIDs, further confounding their classification. Palindromic rheumatism and autoimmune urticaria are temporary rheumatisms and do not fit with the progressive nature of AIDs, and are not included in Figure 2.

### 3.7. Injury-Induced Diseases

Some of the potential AIDs result from injury and are not included in Figure 2. For example, Menière’s disease is an inner ear disorder that can lead to vertigo and hearing loss and is associated with head injury. Ligneous conjunctivitis is another example. Autoimmune retinopathy is an immune response to external injury, and all three are omitted from Figure 2 [98].

Postpericardiotomy syndrome is a subgroup of post-cardiac injury syndromes together with postmyocardial infarction syndrome (Dressler’s syndrome) and posttraumatic pericarditis and are omitted from Figure 2 [99]. 

Giant cell myocarditis (GCM) is not an autoimmune disease in more than 80% of cases [100] and was omitted from Figure 2. 

Parsonage–Turner syndrome, also known as neuralgic amyotrophy, is a poorly understood neuromuscular disorder affecting peripheral nerves mostly within the brachial plexus distribution but can also involve other sites, including the phrenic nerve [101]. The etiology of the syndrome is unclear, and it has been reported in various clinical situations, such as postoperative, postinfectious, posttraumatic, and postvaccination. As such, Parsonage–Turner syndrome is omitted from our list. 

### 3.8. Alzheimer’s Disease Spectrum Includes Autoimmunization

Alzheimer’s disease (AD) is an irredeemable chronic neurodegenerative disorder and the leading cause of dementia in the elderly. Despite rigorous multinational efforts and decades of costly research, no scientific consensus regarding the causes of AD has been achieved. Still, an exclusive pathological event leading to AD development remains a mystery. Growing evidence indicates that AD results from several intertwining pathologies [102,103]. It was also hypothesized that AD possesses an autoimmune component [104,105]; however, little scientific attention has been paid to this theory. However, it was pointed out that AD-associated autoimmunity could be triggered by several self-tolerance and pathogen-related mimicry mechanisms [106], including of bacterial [107], and viral [108] origin.

Several groups tried to provide evidence linking autoimmune diseases with dementia. Wotton and Goldacre performed a retrospective, record-linkage cohort study to determine whether hospital admission for autoimmune disease is associated with an elevated risk of future dementia [109]. Researchers found that 18 different autoimmune diseases, such as lupus, psoriasis, and MS, demonstrated a significant association with dementia. Notably, AD and vascular dementia showed the most significant positive associations with autoimmune pathologies. The authors speculate that the association with vascular dementia is a component of a broader relationship between autoimmune pathologies and neurovascular damage. 

Another recent investigation demonstrated a significant decrease in the total and resting regulatory T cells in AD patients. Surprisingly, a similar phenotype was detected in MS patients. The authors suggest that alterations in regulatory T cells number and activity observed in both diseases play a role in the T cell-mediated immunological tolerance impairment, which indicates a link between these pathologies [110]. Of note, mutual mechanisms for AD and MS were proposed earlier by Avinash Chandra, who hypothesized that amyloid ameliorates disease-associated inflammation [111]. 

Here, we support this hypothesis and show a high association between AD and MS (Figure 2). Moreover, we suggest that AD and MS share common mechanisms of neurodegeneration. The elevation of amyloid precursor protein expression levels in axons around the plaque in MS and the correlation of amyloid-β with distinct stages of multiple sclerosis clearly indicate a role of amyloidosis in MS pathogenesis. 

Undoubtedly, there is primary clinical importance in revealing the putative autoimmune components of AD. A novel complex approach to the disease treatment and diagnosis, which considers several intricate mechanisms, may pave the way to efficient therapy.

## 4. Conclusions

Here, we quantify the association of some 100 different AIDs. The affected systems include: (1) gastrointestinal, (2) neuronal, (3) eye, (4) cutaneous, (5) musculoskeletal, (6) kidneys and lungs, (7) cardiovascular, (8) hematopoietic, (9) endocrine, and (10) multiple. Interestingly, our clustering differentiates between predominantly male and female AIDs. The male/female classification does not explain why most AIDs are more prevalent among women, while others are more prevalent among men. One reason could be the type of tissue affected, and if collagen is targeted, then female predominance could be masked by the high collagen level in male skin. In addition, our clustering of AIDs follows average onset age and systems affected. Our results are expected to assist the diagnosis of comorbidity, and identification of common genetic and environmental factors. 

## Figures and Tables

**Figure 1 jcm-11-04345-f001:**
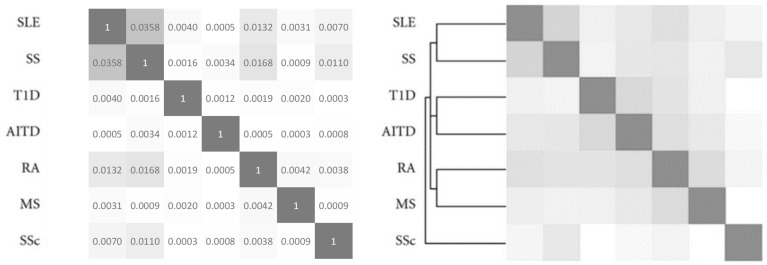
Validation of autoimmune disease association. Shown on the left is our autoimmune disease association matrix based on PubMed co-citation. Shown on the right is the same matrix based on genetic evidence (adapted from Cifuentes et al. [3]). Note that association values range from 0 (white background) to 1 (gray background). Notably, the correlation between the two matrices is high (R = 0.91), thus validating our technique. (SLE, systemic lupus erythematosus. SS, Sjögren’s syndrome. T1D, type 1 diabetes. AITD, autoimmune thyroid disease. RA, rheumatoid arthritis. MS, multiple sclerosis. SSc, systemic sclerosis).

**Figure 2 jcm-11-04345-f002:**
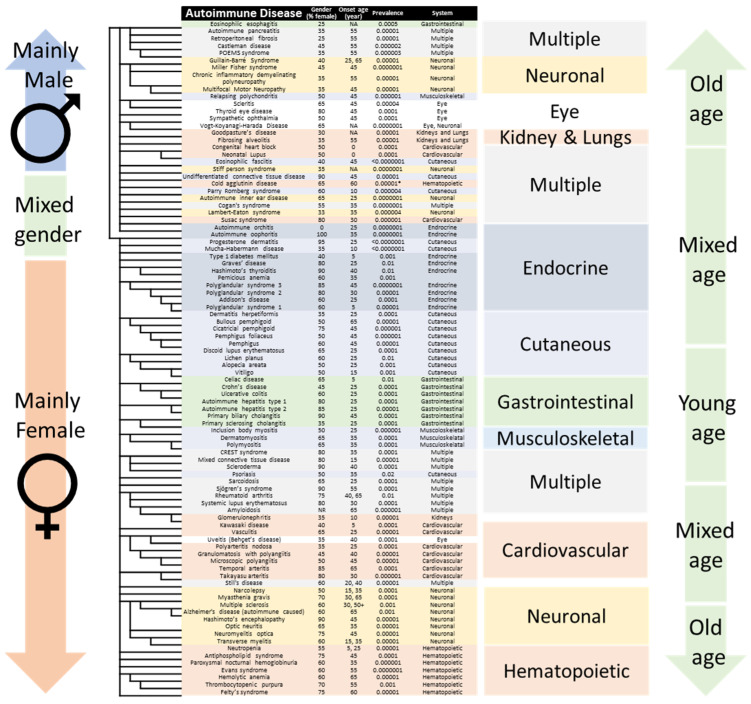
Autoimmune disease classification. Shown is a classification of the top co-cited 100 autoimmune diseases on PubMed. The diseases are clustered using association distances and listed in the order obtained from ClustVis [20]. Notably, the clustering follows gender, system, onset, and prevalence. For a detailed list of the autoimmune diseases, gender ratio, average onset age, prevalence in population, please see Appendix A. (NA–Not applicable, * varies in cold and warm weather).

**Figure 3 jcm-11-04345-f003:**
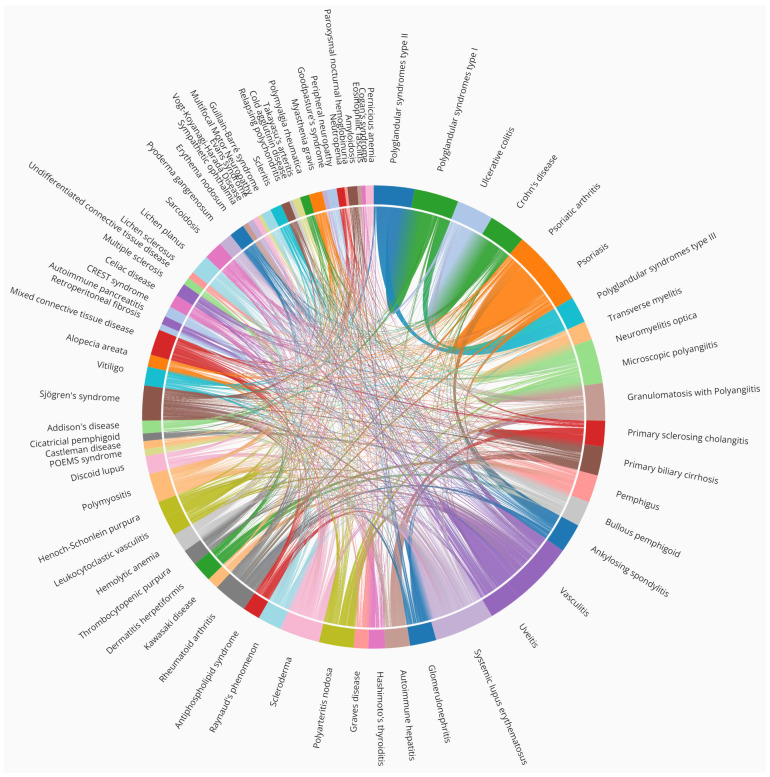
Autoimmune disease associations. Shown is a chord diagram of the top 80 autoimmune diseases based on co-citation in PubMed. The chord width is indicative of the number of co-citations of two AIDs and high comorbidity. The diagram was prepared using Circos software [22].

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
