# Peer review of "Autoimmune Disease Classification Based on PubMed Text Mining"

_jcm, 2022, doi:10.3390/jcm11154345_

Round 1

Reviewer 1 Report

Please punctuate the equations adequately. 

Table 1 is not sufficiently informative to include as such. It would be better as an appendix or supplementary material.

The discussion of the results could be structured in a more informative manner; a higher-level visualization as a weighted graph would be quite useful since Figure 2 has so much detail that is becomes hard to interpret in a fruitful fashion.

The conclusions are a bit on the light side and there is no discussion of future work. 

Author Response

Please find below a point-by-point reply to the reviewers comments.

Reviewer 1

Please punctuate the equations adequately. 

-> We thank the reviewer for this comment, and have corrected the punctuation and explained it as following: “according to the equation below, in which AID_1 and AID_2 correspond to AID pairs:

Association value = (["AID_1" AND "AID_2"])/(["AID_1" OR "AID_2"])            

Table 1 is not sufficiently informative to include as such. It would be better as an appendix or supplementary material.

-> We thank the reviewer for this comment, and have moved Table 1 to supplementary table 2. Instead, a more informative figure has been prepared (see new figure 2 in word file). The new figure legend reads as following:

Figure 2. Autoimmune disease classification. Shown is a classification of the top 100 prevalent autoimmune diseases based on PubMed co-citation. The diseases are clustered using association distances, listed in the order obtained from ClustVis21. Notably, the clustering follows gender, affected system, and average age of onset. (For a detailed list of the autoimmune diseases, gender ratio, average onset age, prevalence in population, please see supplementary table 2).”

The discussion of the results could be structured in a more informative manner; a higher-level visualization as a weighted graph would be quite useful since Figure 2 has so much detail that is becomes hard to interpret in a fruitful fashion.

->  We thank the reviewer for this comment, and have restructured the discussion and result in a more informative manner. In particular, a higher level visualization was introduced in figure 2.

The conclusions are a bit on the light side and there is no discussion of future work.

-> We thank the reviewer for this comment, and have revised the result section extensively and included a discussion.

Reviewer 2 Report

Authors attempt to predict the comorbidity of autoimmune diseases(AIDs) based on their normalized co-citation in PubMed.  Although there are some gaps to fill, to my best understanding they computed an association value for each AID pair using the ratio of the count of PubMed papers where both AIDs are co-occurring to the number of papers where either AID occurs. Hence, an association value of 0 indicates that there is no paper where both AIDs cooccur while an association value of 1 indicates that in all the papers found has both AIDs.

Although predicting potential AID associations based on literature sounds interesting, I think authors should address the following major issues prior to publication:

- The association value will favor rare AIDs. For instance, an association value of 1/1 has different significance then 10000/10000. The impact of such bias needs to be discussed or alternative association value metrics can be tried.

- It is not clear how they search PubMed (e.g. keyword search, MESH term search, or sth else) . It is of little interest to the audience to hear about wget or Python than the query string formation.  A brief description of search approach and its justification would be nice, 

- Papers that discuss autoimmune diseases are typically expected to list some AIDs(like the author's manuscript or a AID review paper), this is not necessarily an indication of association and moreover, such cases will unnecessarily inflate the association between rare AIDs. It will be valuable if or how the impact of this issue is discussed and/or addressed in this manuscript.

- There is a mention of PCA analysis and some results reported, yet I cannot find the details of this analysis; what is passed as input, where does "system affected" come from, and how did they observe gender/system-based separation in PCA? There is no graphical or textual results for PCA analysis

- The methodology for claimed system-affected and gender classification deserves a detailed explanation. As it stands, it is even hard to guess.

- Figure 1 reports significantly very small association values, for a range [0-1]  Is it realistic to argue an association for many? If so, what would be the thresholds 

- It is told; "the list was obtained by downloading several lists of autoimmune diseases". What were the sources of these lists? 

-Pg. 7 to 12 has an extensive overview of the disease. Most of which are textbook material. The underlying value of having this content in this paper is not clear.  This portion certainly increases the number of citations but has little relation, if not none, to the paper's goal.  Or potentially better contextualization is required.

- Figure 2 is illegible, maybe an association value thresholded version would be cleaner. Why did the number of AIDs drop to 100?

- There are mentions of 100 vs. 150 AIDs used. Which one is correct?

- "Alzheimer’s disease spectrum includes autoimmunization." section mentions a high correlation between AD and MS , while the table 1 lacks this information and could not locate any evidence to this point

- What is meant by "non-classical AID" (vs. a classical AID???) ?

Also some minor issues:

- A proofreading will be nice. I spotted few typos; e.g. 

- In the caption Figure 1:  systemic sclerosi.s ->systemic sclerosis

- On Pg.3: Yest -> yet?

Author Response

Please find below a point-by-point reply to the reviewers comments.

Reviewer 2

Authors attempt to predict the comorbidity of autoimmune diseases(AIDs) based on their normalized co-citation in PubMed.  Although there are some gaps to fill, to my best understanding they computed an association value for each AID pair using the ratio of the count of PubMed papers where both AIDs are co-occurring to the number of papers where either AID occurs. Hence, an association value of 0 indicates that there is no paper where both AIDs cooccur while an association value of 1 indicates that in all the papers found has both AIDs.

Although predicting potential AID associations based on literature sounds interesting, I think authors should address the following major issues prior to publication:

- The association value will favor rare AIDs. For instance, an association value of 1/1 has different significance then 10000/10000. The impact of such bias needs to be discussed or alternative association value metrics can be tried.

->We thank the reviewer for this comment. The reviewer correctly states that the nominator of the association values of equation 1 is biased towards prevalent AIDs, because highly cited AIDs give high associations. To counter this bias, the association is divided and normalized by the denominator which favors rare diseases.  This equation is widely used in our group to extract data from PubMed (https://doi.org/10.3390/jcm11082130,  https://doi.org/10.3390/jcm11030781, etc), and provides a good estimate of association.

Association value = (["AID1" AND "AID2"])/(["AID1 " OR "AID2"]),

This is now reflected in the following sentence in the methods section: “The nominator of this equation is biased towards highly cited AIDs. To counter this bias, the association is divided and normalized by the denominator which reduces the association of highly cited AIDs, and increases the association of poorly cited AIDs.”

- It is not clear how they search Pub

Med (e.g. keyword search, MESH term search, or sth else) . It is of little interest to the audience to hear about wget or Python than the query string formation.  A brief description of search approach and its justification would be nice, 

-> We thank the reviewer for this comment and have added the following description in the methods: ”To further reduce the chance of random co-occurrence, we queried only the title, abstract, and other terms of PubMed citations, but not the full text paper. ”

- Papers that discuss autoimmune diseases are typically expected to list some AIDs(like the author's manuscript or a AID review paper), this is not necessarily an indication of association and moreover, such cases will unnecessarily inflate the association between rare AIDs. It will be valuable if or how the impact of this issue is discussed and/or addressed in this manuscript.

-> We thank the reviewer for this comment and have added the following sentence in the methods: “This equation also assumes random noise, and signal to noise is expected to grow by the square root of signal, similar to random walk.”  

- There is a mention of PCA analysis and some results reported, yet I cannot find the details of this analysis; what is passed as input, where does "system affected" come from, and how did they observe gender/system-based separation in PCA? There is no graphical or textual results for PCA analysis

-> We thank the reviewer for this comment, and this is an error. All mentions of “PCA” have been changed to “clustering analysis”. The clustering is shown in the new figure 2.

- The methodology for claimed system-affected and gender classification deserves a detailed explanation. As it stands, it is even hard to guess.

-> We thank the reviewer for this comment, and have added a detailed figure which clearly illustrates the system and gender classification (see new figure 2 in word). The new figure legend reads as following:

Figure 2. Autoimmune disease classification. Shown is a classification of the top 100 prevalent autoimmune diseases based on PubMed co-citation. The diseases are clustered using association distances, listed in the order obtained from ClustVis21. Notably, the clustering follows gender, affected system, and average age of onset. (For a detailed list of the autoimmune diseases, gender ratio, average onset age, prevalence in population, please see supplementary table 2).”

- Figure 1 reports significantly very small association values, for a range [0-1]  Is it realistic to argue an association for many? If so, what would be the thresholds 

-> We thank the reviewer for this comment, and no threshold has been applied, so as not to miss rare associations.

- It is told; "the list was obtained by downloading several lists of autoimmune diseases". What were the sources of these lists? 

-> We thank the reviewer for this comment, and have added the source of our AID lists, in the following sentence: “Autoimmune lists were obtained from the Autoimmune Association website [https://autoimmune.org/disease-information], the Autoimmune Institute website [https://www.autoimmuneinstitute.org/resources/autoimmune-disease-list], and the  Autoimmune Registry [https://www.autoimmuneregistry.org/the-list], and were accessed during 2021.”

-Pg. 7 to 12 has an extensive overview of the disease. Most of which are textbook material. The underlying value of having this content in this paper is not clear.  This portion certainly increases the number of citations but has little relation, if not none, to the paper's goal.  Or potentially better contextualization is required.

-> We thank the reviewer for this comment, and have reviewed p. 7 to 9. We have attempted to contextualize the content to our figures. In addition, the text has been significantly shortened, so as not to seem like text-book material. The changes are too extensive to be included in the response to this comment, and we kindly invite the reviewer to read our revised version.

- Figure 2 is illegible, maybe an association value thresholded version would be cleaner.

-> We thank the reviewer for this comment, and have increased the resolution of the figure.

Why did the number of AIDs drop to 100?

-> We thank the reviewer for this comment, and have added the following clarification: “Inadvertently, some of the 150 diseases were not AIDs, and were omitted from our classification. Furthermore, temporary diseases were also excluded because they do not fit with the progressive nature of AIDs. Finally, indistinct AIDs with low PubMed citation counts were also omitted from our classification. The final list included only 100 AIDs.”  

- There are mentions of 100 vs. 150 AIDs used. Which one is correct?

-> We thank the reviewer for this comment.  Inadvertently, some of the 150 diseases are not AIDs (e.g. Lyme, Chagas, etc), and were omitted from our classification. Furthermore, temporary rheumatisms (i.e. Palindromic rheumatism, autoimmune urticaria, etc) were also excluded because they do not fit with the progressive nature of AIDs. Finally, AIDs diseases with low PubMed citations (i.e. perivenous encephalomyelitis: 14 counts) were also omitted from our classification.  This is now reflected in the following sentence: “Figure 2 shows a classification of the top cited 100 AIDs, discussed in the following paragraph, in light of their association and comorbidity”.

- "Alzheimer’s disease spectrum includes autoimmunization." section mentions a high correlation between AD and MS , while the table 1 lacks this information and could not locate any evidence to this point

->Please see new figure 3.

- What is meant by "non-classical AID" (vs. a classical AID???) ?

-> We thank the reviewer for this comment, and have removed any mention to the biased descriptor “classical”.

Also some minor issues:

- A proofreading will be nice. I spotted few typos; e.g. 

-> We thank the reviewer and have proofread the text in review mode.

- In the caption Figure 1:  systemic sclerosi.s ->systemic sclerosis

-> In accordance with the reviewer request, the word “systemic sclerosi.s” has been changed to “systemic sclerosis yest”.

- On Pg.3: Yest -> yet?

-> In accordance with the reviewer request, the word “yest” has been corrected to “yet”.